# Effects of Microstructure and Texture Evolution on Strength Improvement of an Extruded Mg-10Gd-2Y-0.5Zn-0.3Zr Alloy

**Zhiyong Xue** [1,*], **Xiuzhu Han** [2], **Zhiyong Zhou** [2], **Yanlin Wang** [1,*], **Xuesong Li** [1] **and Jiapeng Wu** [1]

[1]  Institute for Advanced Materials, North China Electric Power University, Beijing 102206, China; lixs@ncepu.edu.cn (X.L.); Wujp@ncepu.edu.cn (J.W.)

[2]  Beijing Institute of Spacecraft System Engineering, Beijing 100094, China; Xiuzhuhan@163.com (X.H.); zyzhou21@sina.com (Z.Z.)

*  Correspondence: xuezy@ncepu.edu.cn (Z.X.); wangyanlin921@aliyun.com (Y.W.);
   Tel./Fax: +86-010-6177-2354 (Z.X. & Y.W.)

**Abstract:** The extrusion process with a large extrusion ratio (36:1) has a great effect on microstructure refinement and strength improvement of the Mg-10Gd-2Y-0.5Zn-0.3Zr alloy. The tensile yield strength, ultimate tensile strength, and elongation of the extruded alloy are 306MPa, 410MPa, and 16.3%, respectively. The causes of strength improvement of the extruded alloy are discussed in detail. The grain refinement is a main strengthening source, contributing ~67MPa to the tensile yield strength of the extruded alloy. Dense precipitation of long period stacking ordered (LPSO) and β′ phases on the matrix and transformation of texture type in the extrusion process also partly increase the strength. In addition, a small number of $\{10\bar{1}2\}$ twins during tensile test is another factor improving the strength of the extruded alloy.

**Keywords:** Mg-10Gd-2Y-0.5Zn-0.3Zr alloy; microstructure; texture evolution; extrusion process; mechanical properties

---

## 1. Introduction

Magnesium alloys show a great potential for lightweight application in aerospace and aviation fields where weight-reduction is of significant importance for cost-saving. However, the relatively low strength of magnesium alloys hinders their wide application. Currently, much attention around the world is focused on how to strengthen magnesium alloys. Recent researches suggest that the addition of rare earth (RE) elements into an Mg alloy is an effective strengthening approach, based on which some Mg-Zn-RE alloys, with excellent mechanical properties at room temperature, have been developed [1–4]. The Mg-Zn-RE alloy is a typical age-hardening alloy due to amounts of strengthening phases precipitating from the matrix during plastic deformation or subsequent ageing treatment [1]. The β″ and β′ are two main strengthening phases in Mg-Zn-RE alloys. The β″ phase with a D019 crystal structure (hexagonal, $a = 0.642$ nm, $c = 0.521$ nm) is a metastable phase precipitating at the early stage of the ageing process, which transforms to the intermediate β′ phase quickly with the increasing of ageing time or temperature [1,2,5]. However, the β′ with a base-centered orthorhombic structure ($a = 0.640$ nm, $b = 2.223$ nm, $c = 0.521$ nm) transforms to the stable and soft $\beta_1$ phase with face-centered cubic (fcc) crystal structure at a higher ageing temperature. Therefore, it is necessary to induce dense precipitation of strengthening β″ and β′ phase on the matrix during plastic deformation or ageing process. Besides, the long period stacking ordered (LPSO) phase, as the dispersed phase, is proposed as another strengthening phase in the Mg-Zn-RE alloy due to its excellent strengthening effect. Its occurrence usually depends on the composition (by addition of Zn) and heat treatment.

The $Mg_{97}Zn_1Y_2$ alloy containing LPSO phase has been developed and exhibits excellent mechanical properties, with a yield strength of 610 MPa and elongation of 5% [3].

In addition, grain refinement is the most primary approach in the strengthening of Mg-Zn-RE alloys, because dynamic recrystallization (DRX) takes place relatively easier during hot working of the Mg alloy compared with other light-weight metal alloys, such as Al alloy and Ti alloy. In production practice, the extrusion process is widely used to refine the microstructure to promote the strength of the Mg-Zn-RE alloy because of the intrinsic advantage of the process [4,6]. Particularly, the extrusion process with a large ratio is able to obviously refine the microstructure and improve the mechanical properties of the Mg-Zn-RE alloy [1,6]. Usually, the strong basal-fiber texture is readily formed in the as-extruded Mg-Zn-RE alloy, causing some grains to rotate to certain orientations, favorable or unfavorable for the activation of basal slips during tensile test [7,8]. Nevertheless, the effects of texture on strength and ductility are so complicated that more deep research should be conducted to clarify the dependence of as-extruded texture on the mechanical properties of the Mg-Zn-RE alloy. The present study primarily analyzed microstructure evolution during the extrusion process and its influence on the mechanical properties of the extruded Mg-10Gd-2Y-0.5Zn-0.3Zr alloy. Through comparing the texture evolution between the initial forged alloy and extruded alloy, the effects of texture on the strength improvement of the extruded alloy are also discussed.

## 2. Experimental

The nominal composition of the initial forged alloy used in this article was Mg-10Gd-2Y-0.5Zn-0.3Zr (wt.%). For the details of casting process and forging process of the alloy refer to the paper [5]. The cylinders, with diameter of 75 mm and height of 20 mm, were cut from the forged billet. The forged cylinders were extruded, with extrusion ratio of 36:1, at the temperature of 345 °C. Extruded rods, with diameter of 12.5 mm and length of ~600 mm, were obtained after cutting the cracked ends. The tensile specimens, with diameter of 5 mm and gauge length of 23 mm, were machined from the rods. The mechanical properties of specimens at room temperature were measured on an INSTRON5559 testing machine (Instron, Norwood, MA, USA) with a crosshead speed of 1 mm/min. The microstructures in the alloy were analyzed by optical microscope (OM) (Shanghai optical instrument factory, Shanghai, China) and scanning electron microscope (SEM) (Zeiss, Dresden, Germany). The specimens were etched in 2% and 5% nitric acid alcohol solution. The characterization of the phases was performed on a JEOL JEM-2100 transmission electron microscope (TEM) (JEOL, Tokyo, Japan), operating at 300kV. The thin foils for TEM were prepared by a ion polishing system. The textures of initial forged alloy and extruded alloy were analyzed by electron back-scattered diffraction pattern analysis (EBSD) with orientation imaging microscopy (OIM; TSL Solutions K.K.) (EDAX, Chicago, IL, USA). EBSD was installed in a field emission scanning electron microscopy and automated EBSD (Zeiss, Dresden, Germany) scanning were performed in the stage control mode with TSL data acquisition software (Version 5.3, EDAX, Mahwah, NJ, USA), on an area of 0.1 $mm^2$ with a step of 0.8 μm. The specimens for the EBSD examination were ground on 800 grit silicon carbide papers and then polished mechanically. Subsequently, the specimens were put into the electrolyte, composed of phosphoric acid and ethyl alcohol (volume ratio 3:5), to remove the strained layer and provide a high quality surface finish. Analysis of the EBSD data was accomplished with TSL OIM analysis software (Version 5.3, EDAX, Mahwah, NJ, USA) and the data, with a confidence index >0.1, were used for texture and twin analysis.

## 3. Results Analysis

### 3.1. Microstructures and Mechanical Properties of the Forged Alloy and Extruded Alloy

Figure 1 shows the microstructure of the forged alloy and extruded alloy. It indicates the microstructure was inhomogeneous in the forged alloy due to the coexistence of some big

and small grains on the matrix, suggesting the occurrence of incomplete dynamic recrystallization (DRX) in the forged alloy, as shown in Figure 1a. Only a few particles with cuboid shape were distributed randomly on the matrix, as shown in Figure 1b. In addition, some lamellae were parallel to each other within individual grain, indicating that the lamellae possess the same orientation in each grain. The average grain size of the forged alloy was about 8 µm. After extrusion deformation, amounts of particles on the matrix and the average grain size was smaller (about 3 µm) than that in the forged alloy, as shown in Figure 1c. The microstructure was relatively homogeneous and the grains were refined obviously, indicating the occurrence of almost complete DRX in the extrusion process. Some lamellae paralleling to each other also could be seen in some grains of the extruded alloy. The amount of precipitates was relatively higher compared with that of the as-forged alloy, due to the high temperature of extrusion, and then some relatively tiny precipitates were formed during the extrusion process. With the increasing of the deformational level during the extrusion process, a large number of smaller particles formed from the alloy precipitates from the alloy. Therefore, the number of precipitates in the extruded alloy was higher compared to that of the as-forged alloy. Figure 2 shows the distribution of grain size analyzed by EBSD. It indicates that the volume fraction of big grains in the forged alloy was higher than that in the extruded alloy, implying the uneven microstructure and incomplete DRX in the forged alloy. The average grain sizes of the forged and extruded alloys were 7.9 µm and 3 µm, respectively, which were consistent with the results measured through OM analysis. Table 1 shows tensile mechanical properties of the forged alloy and extruded alloy, where the strength and elongation of the extruded alloy were obviously higher than that of the forged alloy. The tensile yield strength (TYS), ultimate tensile strength (UTS), and elongation of the extruded alloy were 306 MPa, 410 MPa, and 16.3%, respectively, increasing 96 MPa, 102 MPa, and 8.8% compared with the forged alloy. Therefore, the large extrusion-ratio process shows obvious advantages in promoting the mechanical properties of the Mg alloy.

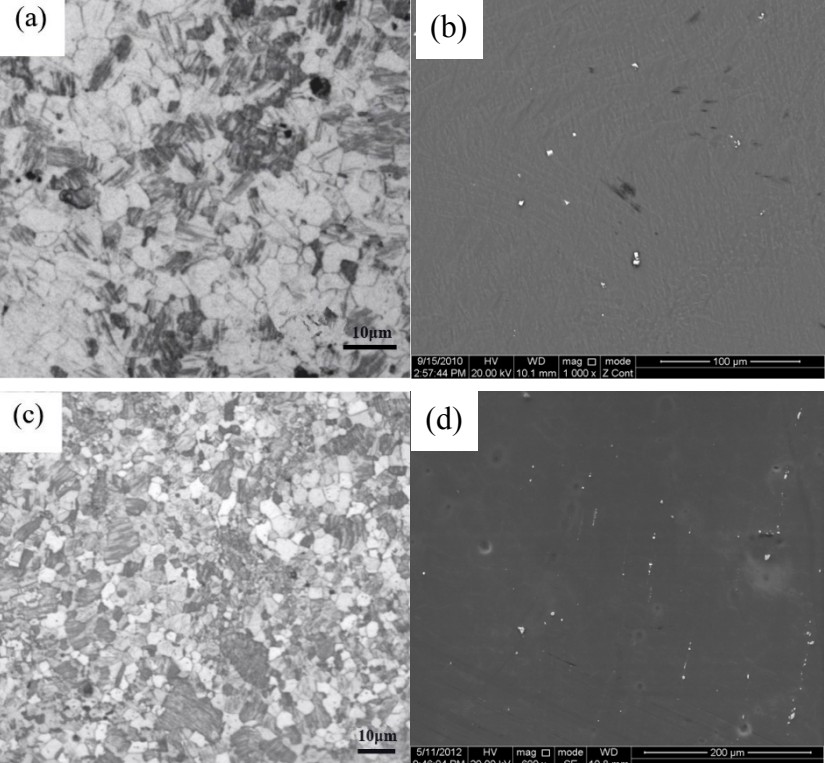

**Figure 1.** Microstructures of the forged alloy and extruded alloy. (**a**,**b**) optical microscope (OM) microstructure and scanning electron microscope (SEM) image of the forged alloy, (**c**) and (**d**) OM microstructure of the transverse section and SEM image of the extruded alloy.

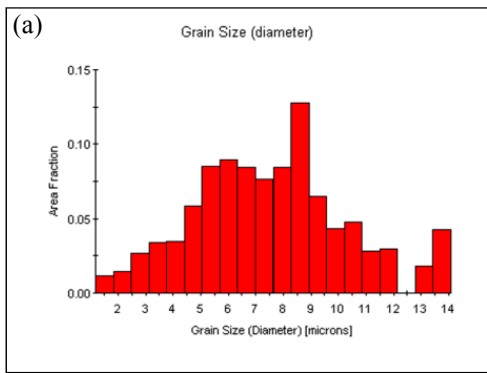
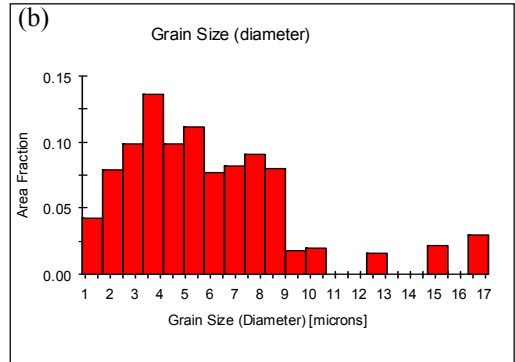

**Figure 2.** The distribution of grain size in the forged alloy and extruded alloy analyzed by electron back-scattered diffraction (EBSD): (**a**) forged alloy; (**b**) extruded alloy.

**Table 1.** Tensile mechanical properties of the forged alloy and extruded alloy.

| State | Tensile Yield Strength (MPa) | Ultimate Tensile Strength (MPa) | Elongation (%) |
|---|---|---|---|
| Forged | 210 | 308 | 7.5 |
| Extruded | 306 | 410 | 16.3 |

*3.2. Precipitates in the Forged Alloy and Extruded Alloy*

Figure 3 shows the TEM images of the second phase particles in the forged alloy and extruded alloy. Amounts of particles, with an average size of about 29.14 ± 3.41 nm, were distributed in a dispersed manner on the matrix in the forged alloy, and the corresponding selected area electron diffraction (SAED) pattern presents some extra diffraction spots at the position of $1/4[01\bar{1}0]_a$, $1/2[01\bar{1}0]a$, and $3/4[01\bar{1}0]a$ Mg reflections, as shown in Figure 3a,b. This suggests the particle is $\beta'$ phase in the forged alloy, which has three variants around $[0001]_\alpha$ zone [4]. Densities of black nanoscale particles, with an average size of about 20.72 ± 2.69 nm, were distributed along the grain boundary in the extruded alloy (indicated by arrows in Figure 3c), and the corresponding SAED pattern (Figure 3d) suggests the particles were also $\beta'$ phase. Actually, the $\beta'$ phase is one of the important strengthening phases in the Mg-RE alloy during plastic deformation and subsequent ageing treatment. The orientation relationships between Mg matrix and $\beta'$ were $[0001]_\alpha \,//\, [001]_{\beta'}$ and $[2\bar{1}\bar{1}0]a \,//\, (100)_{\beta'}$ [9]. Figure 4 presents the lamellar phases in the forged and extruded alloy. A great number of lamellar phases precipitated on the matrix during the extrusion process. The corresponding SAED pattern showed seven extra diffraction spots existing between the central point and $(0001)_\alpha$, confirming that the lamellar phase is the 14H-type LPSO phase. However, some lamellae of the LPSO phase bent at grain boundaries in the extruded alloy, whereas little bending could be observed in the forged alloy. Kinking is an important plastic deformation mode in releasing plastic deformation for the lamellar structure of the metals with less slip systems. Also, it can refine the microstructure and thus increase the strength and ductility of the Mg alloy [4].

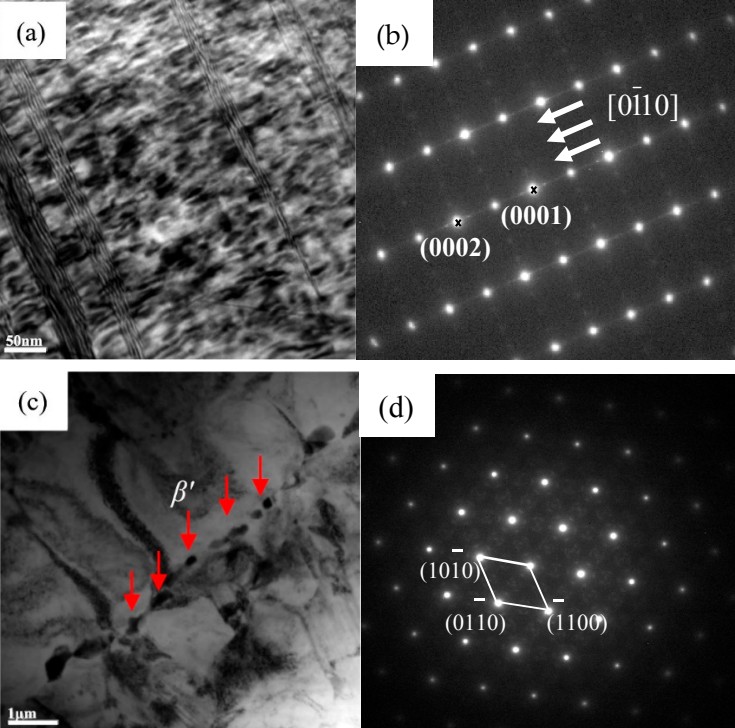

**Figure 3.** (**a**) Bright-field transmission electron microscope (TEM) image and corresponding selected area electron diffraction (SAED) pattern of β′ phase in the forged and extruded alloy; (**b**) TEM image and corresponding SAED pattern of β′ phase in the forged alloy, B // [2$\bar{1}\bar{1}$0]; (**c**) and (**d**) TEM image and corresponding SAED pattern of β′ phase in the extruded alloy, B//[0001].

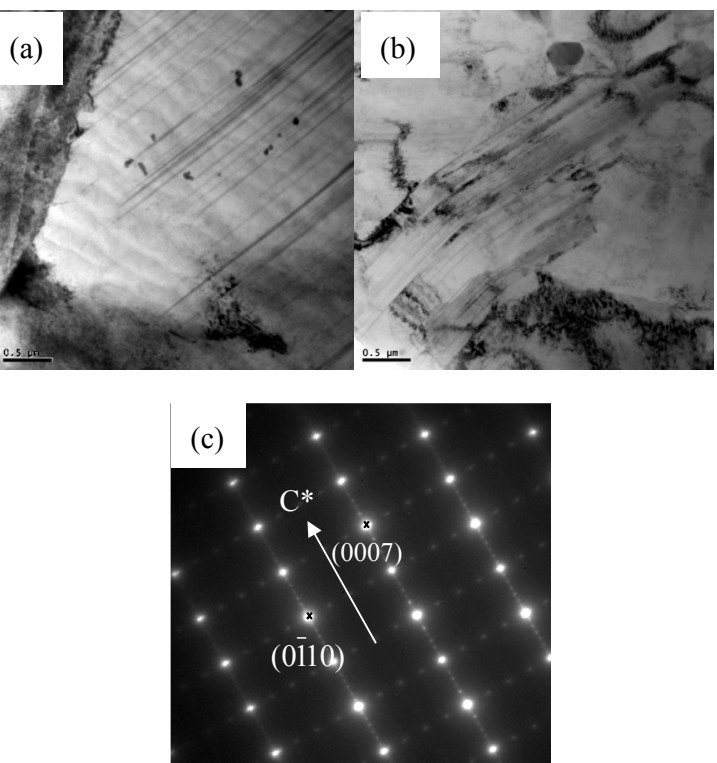

**Figure 4.** TEM image and corresponding SAED pattern of lamellar phase. (**a**) TEM image of lamellar phase in the forged alloy; (**b**) TEM image of lamellar phase in the extruded alloy; and (**c**) SAED pattern of lamellas in (**d**). Electron beam is parallel to [2$\bar{1}\bar{1}$0]*a*.

### 3.3. Texture Evolution in the Forged Alloy and Extruded Alloy

Figure 5 shows the texture evolution in the forged alloy and extruded alloy. It indicates the grains were slightly refined due to DRX in the extruded alloy, as shown in Figure 5a,b. The texture intensity was relatively weaker than that of conventional Mg alloys without RE (10.33 for AZ31 with serve plastic deformation state) [10], mainly attributed to the addition of RE and the randomized texture of recrystallized grains in the alloy. The texture intensity of (0001) in the extruded alloy was a little lower than that of the forged alloy due to the formation of a large number of tiny recrystallized grains in the extrusion process. Furthermore, the c-axis of many grains inclined to an angle to the transverse direction (TD) in the forged alloy, whereas most of those grains disappeared in the extruded alloy. The texture types in the forged alloy and extruded alloy were different from each other, i.e. {0001} <11$\bar{2}$0> in the forged alloy and (0001) basal-fiber texture in the extruded alloy. This suggests the occurrence of grain rotation during the extrusion process, and the effect of grain rotation will be discussed in Section 4.3.

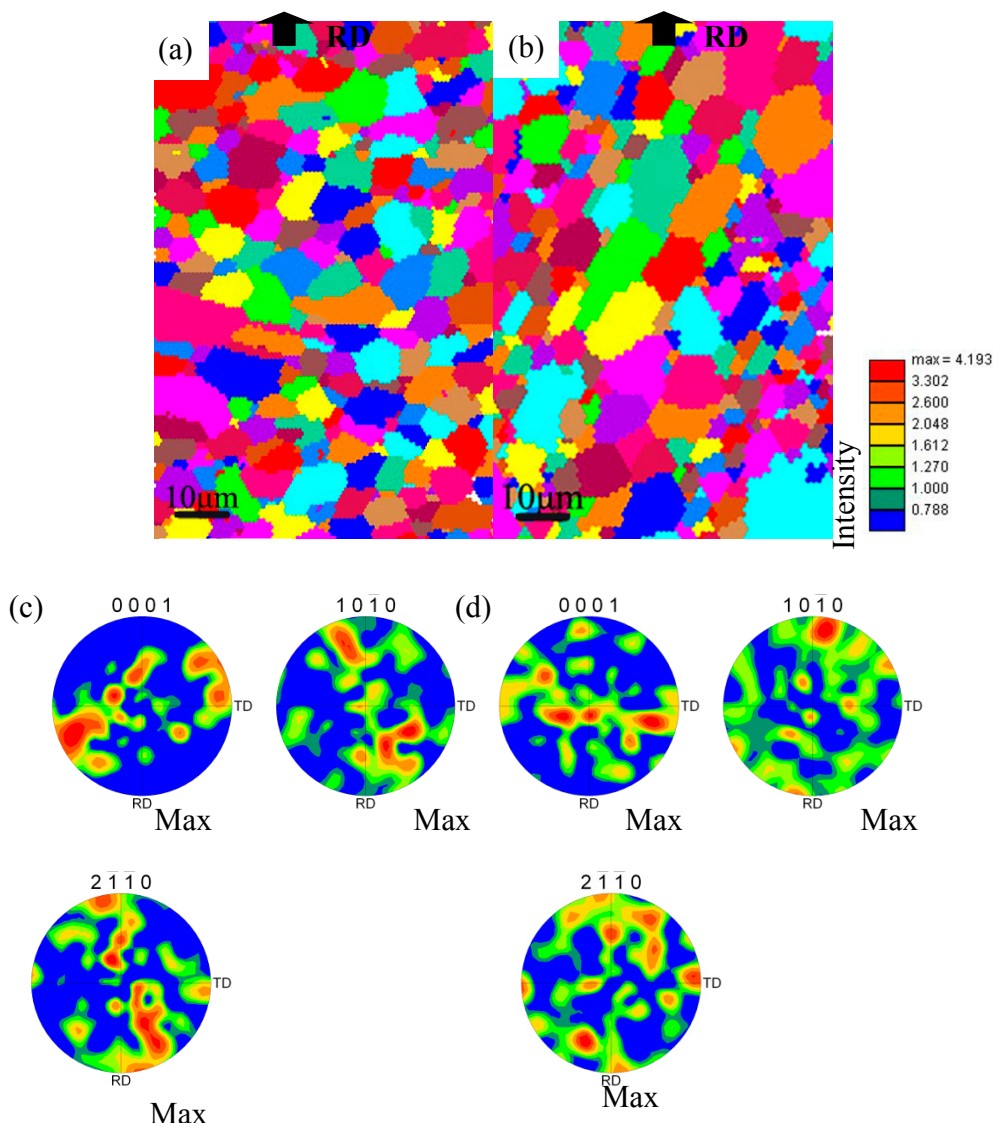

**Figure 5.** Variations of the texture and corresponding (0001),{10$\bar{1}$0}, and{2$\bar{1}\bar{1}$0} pole figures in the forged alloy and extruded alloy. (**a**) and (**c**) forged alloy, (**b**) and (**d**) extruded alloy. RD and ND mean the rolling direction and the normal direction of the sample, respectively, and TD means the transverse direction. The analyzed planes were taken from the plane of RD and TD, which was perpendicular to ND.



## 4. Discussion

### 4.1. Strengthening from Grain Refinement During the Extrusion Process

Grain refinement is an effective method to promote the strength and ductility of the alloys simultaneously. In the present study, the grain is greatly refined in the extrusion process through severe plastic deformation and DRX. The improvement of yield strength due to grain refinement can be estimated by the standard Hall–Petch equation:

$$\Delta\sigma_{\mathrm{gb}} = \frac{k}{\sqrt{d}} \tag{1}$$

where $k$ is the constant related to temperature. The value of $k$ falls between 280~320 MPa $\mu\mathrm{m}^{-1/2}$ in the Mg alloy and, thus the $k$ is 300 MPa $\mu\mathrm{m}^{-1/2}$ chosen in this study [9,11]. Therefore, the strengthening contribution of grain refinement to yield strength improvement can be evaluated by subtracting the $\Delta\sigma_{\mathrm{gb}}$ in the forged alloy from that in the extruded alloy. The increment of yield strength resulting from grain refinement in the extruded alloy is ~67MPa compared with that of the forged alloy. On the other hand, amounts of grain boundaries can effectively inhibit sliding of dislocations and facilitate uniform deformation in tensile tests, resulting in good ductility of the extruded alloy.

### 4.2. Effect of Precipitate on Strength Improvement of the Extruded Alloy

The β′ phase is a main secondary phase both in the forged alloy and extruded alloy. To the best of our knowledge, the β′ phase is one of the important strengthening phases in the Mg-RE alloy, which usually precipitates from the matrix during plastic deformation or subsequent ageing process. In the extruded alloy, the distribution of the β′ phase is much denser than that in the forged alloy, as shown in Figure 3a,c. Furthermore, the tiny β′ phase precipitating from the matrix in the extrusion process is much finer than that of the forged alloy, indicating the obvious microstructure refinement due to large extrusion-ratio plastic deformation. According to the strengthening theory, the nanoscale precipitates have significant effect on precipitation strengthening in the metal alloy [12]. Therefore, the dispersed and nanoscale β′ phase greatly contributes to the strength improvement of the extruded alloy. In addition, the LPSO phase is another important strengthening phase in the Mg-RE alloy, due to its special lamellar morphology obstructing the sliding of dislocations. The densities of the LPSO phases precipitated on the matrix and the kink in the extruded alloy suggests severe plastic deformation during the extrusion process. The kinking can release plastic deformation and avoid stress concentration in local areas, resulting in relatively uniform deformation and better mechanical properties of the extruded alloy. Moreover, the appearance kinking of LPSO lamellae can refine grains more uniformly, which is beneficial to strengthening the extruded alloy. Hence, the comprehensive effects of the dispersed distribution of tiny β′ phase and LPSO phase contribute significantly to the improvement of the strength and ductility of the extruded alloy.

### 4.3. The Role of Texture in Strength Improvement of the Extruded Alloy

Figure 6 shows the different grain orientations and inverse pole figures in the forged and extruded alloy. Texture evolution is another important factor influencing the strength of Mg alloys through changing grain orientation, which may have favorable or unfavorable effects on the strength improvement of the alloy [13,14]. In the present study, the slip of the non-basal plane is ignored due to its high critical resolved shear stress in the tensile test at room temperature. In the forged alloy, although the texture type is (0001)<11$\bar{2}$0>, there are still certain grains whose c-axis inclines ~45° to the RD (calculated from Figure 6c, as indicated in green color region in Figure 6a). In those grains, the slip of the basal planes is relatively easier to activate compared with the ones whose c-axis is perpendicular to the RD in the tensile test. However, in the extruded alloy, the number

of grains with c-axis nearly parallel to ND is increased obviously (indicated in red color in Figure 6b), so the slip systems of basal planes in those grains are difficult to activate in the tensile test, which partly promotes the strength of the extruded alloy. On the other hand, the difficulty in activating basal planes in the extruded alloy also can be seen from the average Schmid factor (SF) for (0001) <$2\overline{11}0$> basal slip, as shown in Table 2. Compared with the forged alloy, the smaller average SF for (0001) basal-fiber texture in the extruded alloy indicates it is more difficult to operate the basal slip system in the tensile test of the extruded alloy. Therefore, the further strength improvement is expected in the extruded alloy relative to the forged alloy.

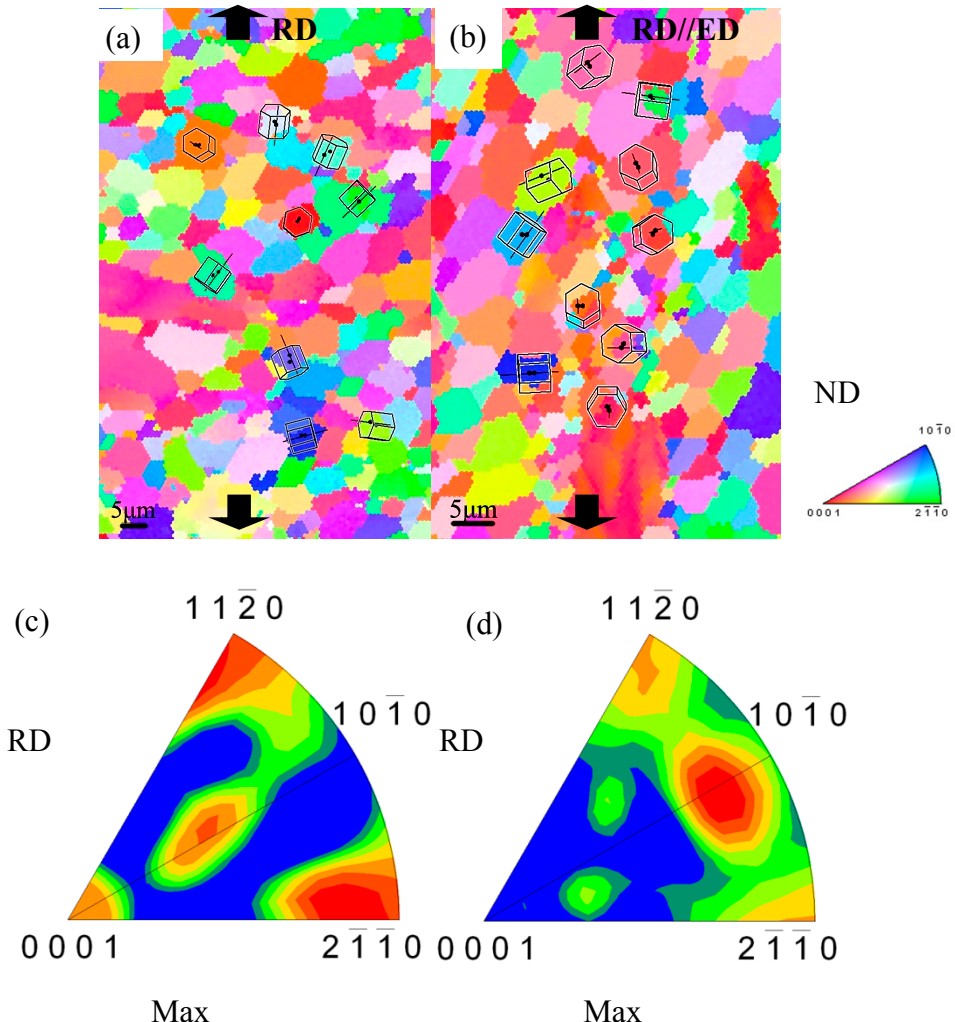

**Figure 6.** (**a**) Different grain orientations and inverse pole figures in the forged and extruded alloy; (**c**) analysis of grain orientation and inverse pole figures in the forged alloy; and (**b**) and (**d**) analysis of grain orientation and inverse pole figures in the extruded alloy. RD and ND mean the rolling direction and the normal direction of the sample, respectively, and TD means the transverse direction. The analyzed planes were taken from the plane of RD and TD, which was perpendicular to ND.

**Table 2.** Average Schmid factors for (0001)<$2\overline{11}0$> basal slip in the extruded alloy.

| Alloy States | Schmid Factor |
| --- | --- |
| Extruded alloy | 0.226 |
| Forged alloy | 0.304 |

### 4.4. The Role of Twinning in Strength Improvement of the Extruded Alloy

According to the results of a previous paper [15], the {10$\bar{1}$2} twinning is easily formed in the grains whose c-axis is parallel to the tensile direction or perpendicular to the compression direction. The formation of the {10$\bar{1}$2} twins can improve the ductility by rotating grains to appropriate directions favoring the activation of the slip systems [16,17]. Therefore, the {10$\bar{1}$2} twinning appears in some certain grains whose c-axis aligns along the RD in the forged alloy, whereas little twinning is formed in the extruded alloy because of lower amount of grains with c-axis parallel to the RD, as shown in Figure 6c,d. Figure 7 shows the distribution of the misorientation angle (the orientation-angle of adjacent grains) in the forged alloy and the extruded alloy, of which samples are taken from the compression test at room temperature. Figure 7a indicates the misorientation angle along the 0° direction and Figure 7b represents that of the 45° direction. One peak, located below 10°, of the misorientation angle appear in the forged alloy, which is a typical characteristic of intersecting or coalescing and nucleation of the {10$\bar{1}$2} twinning in the Mg alloy [15], respectively. Some {10$\bar{1}$2} twins nucleate and grow in tensile tests of the forged alloy, resulting in the increase of the volume fraction of {10$\bar{1}$2} twins, as shown in Figure 7a. Nevertheless, the peak of the misorientation angle below 10° was more evident in the extruded alloy than in the forged alloy, suggesting the occurrence of intersecting or coalescing of {10$\bar{1}$2} twins in the tensile test. Figure 8 shows the twinning characteristics in the extruded alloy. Amounts of {10$\bar{1}$2} twins intersect or coalesce with each other, leading to the reduction of the twins in the extruded alloy. Hence, fewer twinning occurring during tensile tests is another factor for the higher strength of the extruded alloy relative to the forged alloy.

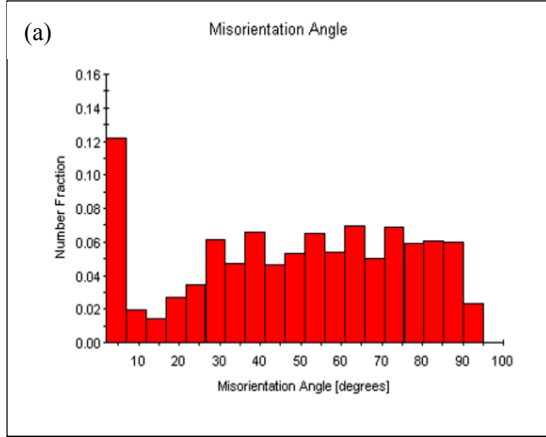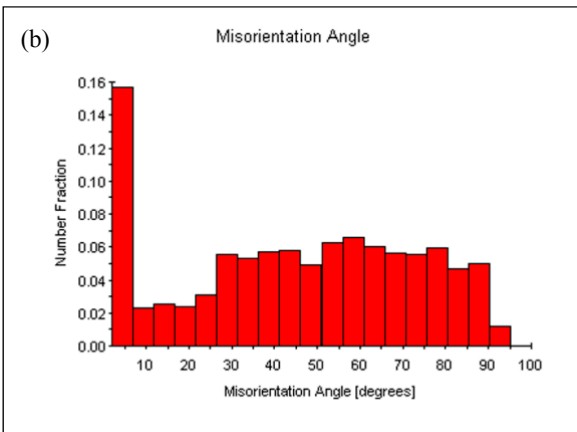

**Figure 7.** Misorientation angle distribution for boundaries in the forged and extruded alloy, compressed at room temperature: (**a**) forged alloy; (**b**) extruded alloy.

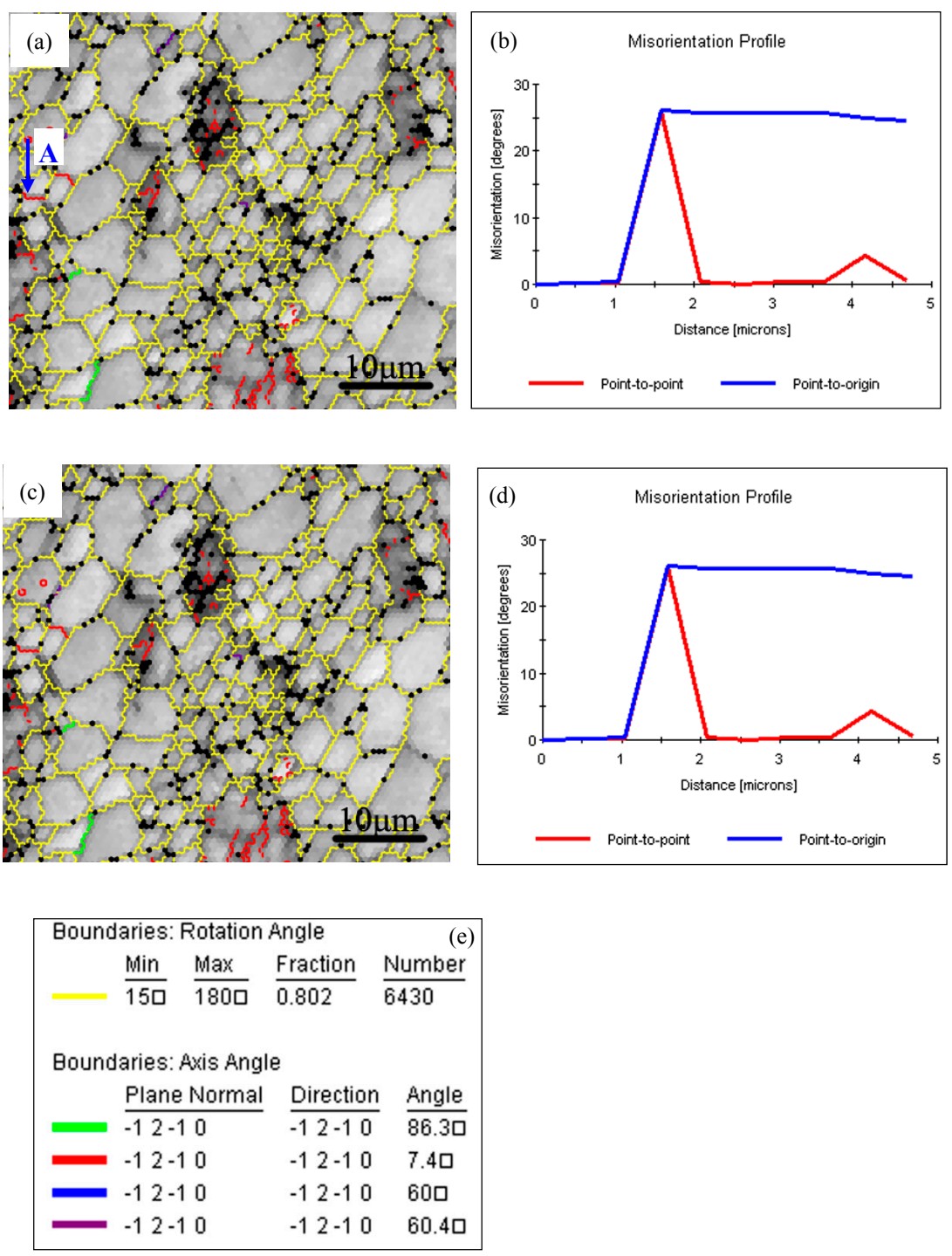

**Figure 8.** Twin characteristics in the extruded alloy compressed at room temperature. (**a**) and (**c**) twining morphology; (**b**) and (**d**) line profiles of the misorientation angle along the direction of arrow A; (**e**) the colors of twins with different misorientation.

## 5. Conclusions

The effects of microstructure and texture evolution on the strength improvement of the as-forged Mg-10Gd-2Y-0.5Zn-0.3Zr alloy in the extrusion process with a high extrusion ratio were studied in the article and some conclusions can be obtained as follows:

(1)　The microstructure is refined and the strength and ductility are obviously improved through the extrusion process with a high extrusion ratio in the extruded alloy. The tensile yield

strength, ultimate tensile strength, and elongation of the extruded alloy are 306 MPa, 410 MPa, and 16.3%, respectively;

(2) Compared with that of the forged alloy, the increment of tensile yield strength of the extruded alloy is ~96MPa, in which ~67MPa arises from grain refinement. Furthermore, the comprehensive effects of dispersed distribution of tiny $\beta'$ phase and LPSO phase also have a large contribution to the strength improvement of the extruded alloy;

(3) The texture transforms from $(0001)<11\bar{2}0>$ in the forged alloy to $(0001)$ basal-fiber texture in the extruded alloy. In the forged alloy, the basal slips in some grains are relatively easier to activate because of their favorable orientations in the tensile test. However, the basal slips are difficult to activate in most of the grains whose c-axis aligns along the normal direction, which is another factor increasing the strength of the extruded alloy.

(4) In the tensile test, some $\{10\bar{1}2\}$ twins are formed in certain grains, with the c-axis parallel to RD in the forged alloy, whereas less nucleation of twins can be found in the extruded alloy. The lower number of $\{10\bar{1}2\}$ twins also partly contribute to the strength improvement of the extruded alloy.

**Author Contributions:** Z.X., X.H. and Y.W. are the main contributor of this research work. They mainly performed the experimental work, results analysis and made draft the research paper; Z.Z., X.L. and J.W. analyzed the data and proofread the paper.

**Funding:** This work is supported by the National Natural Science Foundation of China (No. 51401070), the Fundamental Research Funds for the Central Universities (No. 2014ZZD03) and the China Postdoctoral Science Foundation (2016M591072).

**Conflicts of Interest:** The authors declare no conflict of interest.

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
