# Peer review of "Effects of Microstructure and Texture Evolution on Strength Improvement of an Extruded Mg-10Gd-2Y-0.5Zn-0.3Zr Alloy"

_metals, doi:10.3390/met8121087_

Round 1
Reviewer 1 Report
In this manuscript a study of Mg-10Gd-2Y-0.5Zn-0.3Zr alloy is reported. The alloy is forged, and then extruded. The increase in the strength after extrusion is reported, as well as a detailed study of the change in the microstructure.
This (composition) alloy has been subject of several previous studies. The increase in strength as well as ductility in the present alloy is most likely due to reduction in the grain size. The stress-strain curves, however, are not shown. The microstructural studies are confused (unable to show a clear difference), and conclusions faulty or not convincing. The discussion is inadequate, without any calculations. A detailed report is given below.
Citations are needed for several statements, such as for lines 30-31. Some statements are not correct, such as lines 31-32 [it applies to Mg-RE alloys, and there is no solubility in the matrix]. For the precipitates, citations 1,2 are not adequate. More original citations for these phases are needed.
LPSO is "another strengthening phase", not as precipitates but as dispersed phase, and its occurrence depends on the composition (by addition of Zn). Line 44 "grain refinement is another effective approach" - in fact, grain refinement is the most primary approach in strengthening of metal and alloys. "Particularly, the extrusion process... of Mg-Zn-RE alloy" needs citations.
There is no model of TEM TECNAI-2010. Line 89-90: "particle precipitates" - these are particles (dispersed) or precipitates?
Severe plastic deformation does not occur in extrusion process, and kinking of LPSO phase does not mean that.
Section 4.1: "K falls between 280-320 MPa..." It doesn't seem to be in reference [1] for example. It should be justified how this is chosen, and how much will the YS change based on the average grain size. Discussion in section 4.2 seems meaningless. In Fig. 6 caption one of the '(a) and (b)' should be '(b) and (d).'
Similarly, there is a long list of items that need correction. In reference [3], the first author's name should be 'Kawamura.'
Author Response
Response to the Reviewers’ comments on paper of
metals-381522
“Effects of microstructure and texture evolution on strength improvement of extruded Mg-10Gd-2Y-0.5Zn-0.3Zr alloy”
Dear reviewers:
Thank you very much for your careful review and constructive suggestions with regard to our manuscript “Effects of microstructure and texture evolution on strength improvement of extruded Mg-10Gd-2Y-0.5Zn-0.3Zr alloy”. Those comments are helpful for authors to revise and improve our paper. We have studied comments carefully and tried our best to revise and improve the manuscript and made great changes in the manuscript according to the referees′ good comments. Revised portion is marked in red in the paper. The main corrections in the paper and the responds to the reviewer’s comments are as flowing. We appreciate for Editors/Reviewers’ warm work earnestly, and hope that the corrections will meet with approval. Please feel free to contact us with any questions and we are looking forward to your consideration. We would like to express our great appreciation to you and reviewers for comments on our paper. Looking forward to hearing from you.
Thank you and best regards.
Yours sincerely.
Review 1#
Comments and Suggestions for Authors
In this manuscript a study of Mg-10Gd-2Y-0.5Zn-0.3Zr alloy is reported. The alloy is forged, and then extruded. The increase in the strength after extrusion is reported, as well as a detailed study of the change in the microstructure.
(1) This (composition) alloy has been subject of several previous studies. The increase in strength as well as ductility in the present alloy is most likely due to reduction in the grain size. The stress-strain curves, however, are not shown. The microstructural studies are confused (unable to show a clear difference), and conclusions faulty or not convincing. The discussion is inadequate, without any calculations. A detailed report is given below.
Answer 1: Thank you for your suggestion, in this revised version, we have made a correction according to the reviewer’s advice, and the corresponding corrections have been made in red font in the revised manuscript.
(2) Citations are needed for several statements, such as for lines 30-31. Some statements are not correct, such as lines 31-32 [it applies to Mg-RE alloys, and there is no solubility in the matrix]. For the precipitates, citations 1,2 are not adequate. More original citations for these phases are needed.
Answer 2: Thank you for your suggestion, we have made a correction according to the reviewer’s advice, and the corresponding corrections have been made in red font in the revised manuscript.
(3) LPSO is "another strengthening phase", not as precipitates but as dispersed phase, and its occurrence depends on the composition (by addition of Zn). Line 44 "grain refinement is another effective approach" - in fact, grain refinement is the most primary approach in strengthening of metal and alloys. "Particularly, the extrusion process... of Mg-Zn-RE alloy" needs citations.
Answer 3: Your opinion very correct, thank you, we have made a correction according to the reviewer’s advice, and the corresponding corrections have been made in red font in the revised manuscript.
(4 ) There is no model of TEM TECNAI-2010. Line 89-90: "particle precipitates" - these are particles (dispersed) or precipitates?
Answer 4: Thank you for your suggestion, figure.3 shows the TEM images of the second phase particles in the forged alloy and extruded alloy, amounts of particles are dispersively distributed on the matrix in the forged alloy in this revised version.
(5) Severe plastic deformation does not occur in extrusion process, and kinking of LPSO phase does not mean that.
Answer 5: Thank you for your suggestion, yes, severe plastic deformation does not occur in extrusion process, in this revised version, we have made a correction according to the reviewer’s advice.
(6) Section 4.1: "K falls between 280-320 MPa..." It doesn't seem to be in reference [1] for example. It should be justified how this is chosen, and how much will the YS change based on the average grain size. Discussion in section 4.2 seems meaningless. In Fig. 6 caption one of the '(a) and (b)' should be '(b) and (d).' Similarly, there is a long list of items that need correction. In reference [3], the first author's name should be 'Kawamura.'
Answer 6: Thank you for your suggestion, we have made a correction according to the reviewer’s advice, and the corresponding corrections have been made in red font in the revised manuscript.
All the revisions are made in highlighted font so that the reviewer can easily view the changes. If there is any problem, please do not hesitate to contact me by any of the following methods. Thank you very much.

Reviewer 2 Report
This work tries to clarify the strength improvement of a Mg-Gd based alloy by an extrusion. The reviewer agrees that the grain refinement is important to enhance both strength and ductility; however, to submit this work as a full paper article, the authors had better to care about following things.
1) The reviewer could not see the β’ precipitates in Fig.3c. From this image, we cannot discuss about their size and number density, and hence, it is difficult for the reviewer to follow the idea that the β’ precipitates contribute the high strengths. The authors are advised to provide STEM image, or should consider to get another image.
2) What are the maps presented in Fig. 5? The grains have several colors, but the reviewer cannot not understand the meaning of colors. From these images, the reviewer thinks that the grain size of the extruded material is almost the same as the forged one. If the authors use different magnification, it’s good to use the same magnification. In addition, the grain size distribution, Table 2, should be provided along with these maps because it seems that the grain size is calculated from the EBSD.
3) The authors provide the Fig. 6 without any detailed explanations. So the reviewer cannot not follow what the authors want to emphasize.
4) The authors might want to mention the importance of twinning in Fig. 7 and 8. Are they obtained after some straining of tensile test? Detail explanations are missing, please state how the authors obtain these graphs and map.
Minor points
5) What the RD,TD, and ND mean in Fig. 5 and Fig. 6? And what planes are analyzed by EBSD?
6) The “K” in page 7, line 169 should be “k”.
7) The “Fig. 1” in page 7, line 180 seems to be “Fig. 3”.
Author Response
Response to the Reviewers’ comments on paper of
metals-381522
“Effects of microstructure and texture evolution on strength improvement of extruded Mg-10Gd-2Y-0.5Zn-0.3Zr alloy”
Dear reviewers:
Thank you very much for your careful review and constructive suggestions with regard to our manuscript “Effects of microstructure and texture evolution on strength improvement of extruded Mg-10Gd-2Y-0.5Zn-0.3Zr alloy”. Those comments are helpful for authors to revise and improve our paper. We have studied comments carefully and tried our best to revise and improve the manuscript and made great changes in the manuscript according to the referees′ good comments. Revised portion is marked in red in the paper. The main corrections in the paper and the responds to the reviewer’s comments are as flowing. We appreciate for Editors/Reviewers’ warm work earnestly, and hope that the corrections will meet with approval. Please feel free to contact us with any questions and we are looking forward to your consideration. We would like to express our great appreciation to you and reviewers for comments on our paper. Looking forward to hearing from you.
Thank you and best regards.
Yours sincerely.
Review 2#
This work tries to clarify the strength improvement of a Mg-Gd based alloy by an extrusion. The reviewer agrees that the grain refinement is important to enhance both strength and ductility; however, to submit this work as a full paper article, the authors had better to care about following things.
1) The reviewer could not see the β’ precipitates in Fig.3c. From this image, we cannot discuss about their size and number density, and hence, it is difficult for the reviewer to follow the idea that the β’ precipitates contribute the high strengths. The authors are advised to provide STEM image, or should consider to get another image.
Answer 1: Thank you for your suggestion, in this revised version, we have made a correction according to the reviewer’s advice.
2) What are the maps presented in Fig. 5? The grains have several colors, but the reviewer cannot not understand the meaning of colors. From these images, the reviewer thinks that the grain size of the extruded material is almost the same as the forged one. If the authors use different magnification, it’s good to use the same magnification. In addition, the grain size distribution, Table 2, should be provided along with these maps because it seems that the grain size is calculated from the EBSD.
Answer 2: In this revised version, we have made a correction according to the reviewer’s advice, and the corresponding corrections have been made in red font in the revised manuscript.
3) The authors provide the Fig. 6 without any detailed explanations. So the reviewer cannot not follow what the authors want to emphasize.
Answer 3: Thank you for your suggestion, in this revised version, we have added the corresponding informations according to the your advice, and the corresponding corrections have been made in red font in the revised manuscript.
4) The authors might want to mention the importance of twinning in Fig. 7 and 8. Are they obtained after some straining of tensile test? Detail explanations are missing, please state how the authors obtain these graphs and map.
Answer 4: In this revised version, we have made a correction according to the reviewer’s advice.
5) What the RD,TD, and ND mean in Fig. 5 and Fig. 6? And what planes are analyzed by EBSD?
Answer 5: Thank you for your suggestion, the corresponding corrections have been made in red font in the revised manuscript.
6) The “K” in page 7, line 169 should be “k”.
Answer 6: Thank you for your suggestion, we have made a correction according to the reviewer’s advice, the corresponding corrections have been made in red font in the revised manuscript.
7) The “Fig. 1” in page 7, line 180 seems to be “Fig. 3”.
Answer 7: Thank you for your suggestion, the corresponding corrections have been made in red font in the revised manuscript.
All the revisions are made in highlighted font so that the reviewer can easily view the changes. If there is any problem, please do not hesitate to contact me by any of the following methods. Thank you very much.

Reviewer 3 Report
This manuscript presents some observations relating to the effects of microstructure and texture on strength properties of Mg-based alloys in the forged and extruded conditions. The topic is interesting for many industrial applications. In general, the paper is well organized and due reference is made to previous work. I have a few comments that the authors are kindly requested to consider in order to further improve their manuscript, which is already of good quality.
1) The syntax and grammar need some revision. In particular the sentence in lines 186-188 is difficult to understand. Maybe one verb is missing? Also things like “increasing of extrusion process” (line 96) or “because of less certain grains” (line 224), need some processing. In the last case, maybe the authors meant “due to the lower amount of grains with c-axis parallel to the RD”?
2) In Figure 1, the magnification for (a) and (c) is not the same. This has the unfortunate effect of artificially increasing the relative apparent grain size difference. It is recommendable to use the same magnification for both images. Maybe also for Fig.1 (b) and (d), for which there is a factor of 2 of difference.
3) Line 124: reference [9] should not appear as a superscript.
4) In figure 3(b), there are two missing “)”: (0001) and (0002) instead of (0001 and (0002.
5) Also in figure 3 (a) and (c), the scale bar and size label are very difficult to see due to the dark background. What about using white color? There is also kind of a scale bar in Fig.3(d). Is this normal?
6) In lines 146-148, a comparison is made with the texture intensity of RE-free Mg alloys. It would be good to mention a value and a reference.
7) Fig.5 (c) and (d): what is the standard deviation on the Max. values? Are these differences significant? The number of grains is relatively low. Are the Max. values reproducible on other areas of the same samples?
8) On lines 169, the constant k is written K. Please the same notation for consistency.
9) In the caption of figure 6, both the forged and extruded alloys are referred to (a) and (c). One of those (extruded probably) should be (b) and (d).
10) Line 225: although it is in principle clear, it would be good to indicate what is the misorientation angle relative to for the non-specialists.
11) Figure 7(a) and related text: there is no real peak between 80 and 90 degrees. This should be rephrased.
12) Figures 8(a) and (b): I am wondering if the blue arrow in 8(a) corresponds precisely to the data in 8(b), because the grain size in this area is such that the first grain boundary should not have been encountered before 5 micrometers.
Author Response
Response to the Reviewers’ comments on paper of
metals-381522
“Effects of microstructure and texture evolution on strength improvement of extruded Mg-10Gd-2Y-0.5Zn-0.3Zr alloy”
Dear reviewers:
Thank you very much for your careful review and constructive suggestions with regard to our manuscript “Effects of microstructure and texture evolution on strength improvement of extruded Mg-10Gd-2Y-0.5Zn-0.3Zr alloy”. Those comments are helpful for authors to revise and improve our paper. We have studied comments carefully and tried our best to revise and improve the manuscript and made great changes in the manuscript according to the referees′ good comments. Revised portion is marked in red in the paper. The main corrections in the paper and the responds to the reviewer’s comments are as flowing. We appreciate for Editors/Reviewers’ warm work earnestly, and hope that the corrections will meet with approval. Please feel free to contact us with any questions and we are looking forward to your consideration. We would like to express our great appreciation to you and reviewers for comments on our paper. Looking forward to hearing from you.
Thank you and best regards.
Yours sincerely.
Review 3#
This manuscript presents some observations relating to the effects of microstructure and texture on strength properties of Mg-based alloys in the forged and extruded conditions. The topic is interesting for many industrial applications. In general, the paper is well organized and due reference is made to previous work. I have a few comments that the authors are kindly requested to consider in order to further improve their manuscript, which is already of good quality.
1) The syntax and grammar need some revision. In particular the sentence in lines 186-188 is difficult to understand. Maybe one verb is missing? Also things like “increasing of extrusion process” (line 96) or “because of less certain grains” (line 224), need some processing. In the last case, maybe the authors meant “due to the lower amount of grains with c-axis parallel to the RD”?
Answer 1: Thank you for your suggestion, we have made a correction according to the reviewer’s advice, the corresponding corrections have been made in red font in the revised manuscript.
2) In Figure 1, the magnification for (a) and (c) is not the same. This has the unfortunate effect of artificially increasing the relative apparent grain size difference. It is recommendable to use the same magnification for both images. Maybe also for Fig.1 (b) and (d), for which there is a factor of 2 of difference.
Answer 2: Thank you for your suggestion, in this revised version, we have made a correction according to the reviewer’s advice.
3) Line 124: reference [9] should not appear as a superscript.
Answer 3: Thank you for your suggestion, in this revised version, we have made a correction according to the reviewer’s advice.
4) In figure 3(b), there are two missing “)”: (0001) and (0002) instead of (0001 and (0002.
Answer 4: Thank you for your suggestion, in this revised version, we have made a correction according to the reviewer’s advice.
5) Also in figure 3 (a) and (c), the scale bar and size label are very difficult to see due to the dark background. What about using white color? There is also kind of a scale bar in Fig.3(d). Is this normal?
Answer 5: We have tried our best to make adjustments, in this revised version, we have made a correction according to the reviewer’s advice.
6) In lines 146-148, a comparison is made with the texture intensity of RE-free Mg alloys. It would be good to mention a value and a reference.
Answer 6: Thank you for your suggestion, in this revised version, we have made a correction according to the reviewer’s advice.
7) Fig.5 (c) and (d): what is the standard deviation on the Max. values? Are these differences significant? The number of grains is relatively low. Are the Max. values reproducible on other areas of the same samples?
Answer 7: The Max. Values is 5.2, and these differences are significant, and the number of grains is relatively low, in this revised version, we have made a correction according to the reviewer’s advice.
8) On lines 169, the constant k is written K. Please the same notation for consistency.
Answer 8: In this revised version, we have made a correction according to the reviewer’s advice.
9) In the caption of figure 6, both the forged and extruded alloys are referred to (a) and (c). One of those (extruded probably) should be (b) and (d).
Answer 9: Thank you for your suggestion, in this revised version, we have made a correction according to the reviewer’s advice.
10) Line 225: although it is in principle clear, it would be good to indicate what is the misorientation angle relative to for the non-specialists.
Answer 10:Thank you for your suggestion, in this revised version, we have made a correction according to the reviewer’s advice.
11) Figure 7(a) and related text: there is no real peak between 80 and 90 degrees. This should be rephrased.
Answer 11:Thank you for your suggestion, in this revised version, we have made a correction according to the reviewer’s advice.
12) Figures 8(a) and (b): I am wondering if the blue arrow in 8(a) corresponds precisely to the data in 8(b), because the grain size in this area is such that the first grain boundary should not have been encountered before 5 micrometers.
Answer 12: Thank you for your suggestion, the blue arrow in 8(a) corresponds precisely to the data in 8(b), in this revised version, we have made a correction according to the reviewer’s advice.
All the revisions are made in highlighted font so that the reviewer can easily view the changes. If there is any problem, please do not hesitate to contact me by any of the following methods. Thank you very much.

Round 2
Reviewer 1 Report
I am still unable to recommend this manuscript for publication because of the poor quality of data and presentation, as well as analysis and language.
Author Response
Dear reviewers and editors, based on your second opinion, we have re-edited on the basis of the first reply, due to the long modification time, I'm sorry, thank you again!
Response to the Reviewers’ comments on paper of
metals-381522
“Effects of microstructure and texture evolution on strength improvement of extruded Mg-10Gd-2Y-0.5Zn-0.3Zr alloy”
Dear reviewers:
Thank you very much for your careful review and constructive suggestions with regard to our manuscript “Effects of microstructure and texture evolution on strength improvement of extruded Mg-10Gd-2Y-0.5Zn-0.3Zr alloy”. Those comments are helpful for authors to revise and improve our paper. We have studied comments carefully and tried our best to revise and improve the manuscript and made great changes in the manuscript according to the referees′ good comments. Revised portion is marked in red in the paper. The main corrections in the paper and the responds to the reviewer’s comments are as flowing. We appreciate for Editors/Reviewers’ warm work earnestly, and hope that the corrections will meet with approval. Please feel free to contact us with any questions and we are looking forward to your consideration. We would like to express our great appreciation to you and reviewers for comments on our paper. Looking forward to hearing from you.
Thank you and best regards.
Yours sincerely.
Review 1#
In this manuscript a study of Mg-10Gd-2Y-0.5Zn-0.3Zr alloy is reported. The alloy is forged, and then extruded. The increase in the strength after extrusion is reported, as well as a detailed study of the change in the microstructure.
This (composition) alloy has been subject of several previous studies. The increase in strength as well as ductility in the present alloy is most likely due to reduction in the grain size. The stress-strain curves, however, are not shown. The microstructural studies are confused (unable to show a clear difference), and conclusions faulty or not convincing. The discussion is inadequate, without any calculations. A detailed report is given below.
1) Citations are needed for several statements, such as for lines 30-31. Some statements are not correct, such as lines 31-32 [it applies to Mg-RE alloys, and there is no solubility in the matrix]. For the precipitates, citations 1,2 are not adequate. More original citations for these phases are needed.
Answer 1: Thank you for your suggestion. We have added some references for some corresponding statements in the 1st paragraph in the Introduction section. We also deleted the incorrect statement “large solid solution of RE elements in Mg matrix and”. (the 7th line in the 1st paragraph in the Introduction section of the revised manuscript).
2) LPSO is "another strengthening phase", not as precipitates but as dispersed phase, and its occurrence depends on the composition (by addition of Zn). Line 44 "grain refinement is another effective approach" - in fact, grain refinement is the most primary approach in strengthening of metal and alloys. "Particularly, the extrusion process... of Mg-Zn-RE alloy" needs citations.
Answer 2: Thank you for your suggestion. We added some description to state the nature of the LPSO phase (the 17th line in the 1st paragraph in the Introduction section of the revised manuscript). We revised the statement about “grain refinement” (the 1st line in the 2nd paragraph in the Introduction section of the revised manuscript). We also added two references for the statement of “Particularly, the extrusion process... of Mg-Zn-RE alloy”. (the 8th line in the 2nd paragraph in the Introduction section of the revised manuscript).
3) There is no model of TEM TECNAI-2010. Line 89-90: "particle precipitates" - these are particles (dispersed) or precipitates?
Answer 3: Thank you for your suggestion. We have corrected the TEM model. (the 11th line in the 1st paragraph in the Experimental section of the revised manuscript) We deleted the word “precipitate” to avoid ambiguity (the 8th line in the 1st paragraph in the section 3.1 of the revised manuscript).
4) Severe plastic deformation does not occur in extrusion process, and kinking of LPSO phase does not mean that.
Answer 4: We accept your point of view and have revised the corresponding sentence. (the 15th line in the 1st paragraph in the section 3.2 of the revised manuscript)
5) Section 4.1: "K falls between 280-320 MPa..." It doesn't seem to be in reference [1] for example. It should be justified how this is chosen, and how much will the YS change based on the average grain size. Discussion in section 4.2 seems meaningless. In Fig. 6 caption one of the '(a) and (b)' should be '(b) and (d).'
Answer 5: Thank you for your suggestion. k = 300 MPa μm−1/2 in this study, this date is chosen based on the main compositions of alloy and we have provided two references ([9] and [10]). Section 4.2 mainly talks about the precipitates strengthening effect, due to containing rich amount of β′ phase in this studied alloy. We also corrected (a) and (c) in Fig.6 to (b) and (d).
6) Similarly, there is a long list of items that need correction. In reference [3], the first author's name should be 'Kawamura.'
Answer 6: Thank you for your suggestion. We have inspected and revised the authors’ name and the serial number for all references in the References section.
All the revisions are made in highlighted font so that the reviewer can easily view the changes. If there is any problem, please do not hesitate to contact me by any of the following methods. Thank you very much.
Reviewer 2 Report
1) The authors conclude that the tiny b’ phases significantly contribute the high strengths in the extruded alloy. As an evidence, the authors provide the TEM images in Fig.3; however, the size of the b’ phases in the extruded sample is quite larger than those in the forged one, so the reviewer cannot believe the author’s opinion. This problem might come from the difference of the magnification of two TEM images. The authors should take the TEM images at higher magnification for the extruded alloy. Probably, much finer precipitates exist in the matrix. If the authors cannot find the finer precipitates, it is difficult to support the author’s idea that the strength is significantly increased due to the fine precipitates in the extruded alloy.
2) Is the grain size really refined by the extrusion process? As the reviewer pointed out previously, the maps presented in the Fig.5 give the wrong impression that the grain size is almost the same for both samples. In addition, the reviewer find that the length of the scale bars in the OM images (Fig. 1) are different. Why the authors avoid using the same magnification? If the authors would like to discuss the change of grain size, providing images with same magnification is essential.
Author Response
Response to the Reviewers’ comments on paper of
metals-381522
“Effects of microstructure and texture evolution on strength improvement of extruded Mg-10Gd-2Y-0.5Zn-0.3Zr alloy”
Dear reviewers:
Thank you very much for your careful review and constructive suggestions with regard to our manuscript “Effects of microstructure and texture evolution on strength improvement of extruded Mg-10Gd-2Y-0.5Zn-0.3Zr alloy”. Those comments are helpful for authors to revise and improve our paper. We have studied comments carefully and tried our best to revise and improve the manuscript and made great changes in the manuscript according to the referees′ good comments. Revised portion is marked in red in the paper. The main corrections in the paper and the responds to the reviewer’s comments are as flowing. We appreciate for Editors/Reviewers’ warm work earnestly, and hope that the corrections will meet with approval. Please feel free to contact us with any questions and we are looking forward to your consideration. We would like to express our great appreciation to you and reviewers for comments on our paper. Looking forward to hearing from you.
Thank you and best regards.
Yours sincerely.
Review 2#
This work tries to clarify the strength improvement of a Mg-Gd based alloy by an extrusion. The reviewer agrees that the grain refinement is important to enhance both strength and ductility; however, to submit this work as a full paper article, the authors had better to care about following things.
1) The reviewer could not see the β’ precipitates in Fig.3c. From this image, we cannot discuss about their size and number density, and hence, it is difficult for the reviewer to follow the idea that the β’ precipitates contribute the high strengths. The authors are advised to provide STEM image, or should consider to get another image.
Answer 1: Thank you for your suggestion. We have added some arrows in Fig.3 (c) to indicate the β′ phase in the revised manuscript. We hope that this revision will help you and readers better understand the relevant descriptions in the paper. Additionally, the SAED pattern shown in Fig. 3(d) has been a good proof that those black nonalscale particles shown in Fig.3 (c) are β′ phase. So we don’t think it necessary to replace this TEM image.
2) What are the maps presented in Fig. 5? The grains have several colors, but the reviewer cannot not understand the meaning of colors. From these images, the reviewer thinks that the grain size of the extruded material is almost the same as the forged one. If the authors use different magnification, it’s good to use the same magnification. In addition, the grain size distribution, Table 2, should be provided along with these maps because it seems that the grain size is calculated from the EBSD.
Answer 2: We have added a map in Fig. 5 to understand the meaning of colors. After a closer look, we agree with your opinion, namely, the grain size difference between the two states (extruded and forged) is not large, and we have modified the relevant expressions in this paper.
3) The authors provide the Fig. 6 without any detailed explanations. So the reviewer cannot not follow what the authors want to emphasize.
Answer 3: Thank you for your suggestion. According to your comment, we added some explanations about Fig. 6 to make the description of this figure clearer.
4) The authors might want to mention the importance of twinning in Fig. 7 and 8. Are they obtained after some straining of tensile test? Detail explanations are missing, please state how the authors obtain these graphs and map.
Answer 4: These twins form during compression test conducted at room temperature. We have added the relevant descriptions in the revised manuscript.
5) What the RD,TD, and ND mean in Fig. 5 and Fig. 6? And what planes are analyzed by EBSD?
Answer 5: RD and ND mean that the rolling (or extruding) direction and the normal direction of the sample, respectively, and TD means the transverse direction. The analyzed planes were taken from the plane of RD and TD, which was perpendicular to ND. We have added a clear description in the revised manuscript.
6) The “K” in page 7, line 169 should be “k”.
Answer 6: Thank you for your suggestion, we have made a correction according to your advice (The 2nd line of the 2nd paragraph in the section 4.1).
7) The “Fig. 1” in page 7, line 180 seems to be “Fig. 3”.
Answer 7: Thank you for your suggestion, we have made a correction according to your advice (The 5th line of the 1st paragraph in the section 4.2)
All the revisions are made in highlighted font so that the reviewer can easily view the changes. If there is any problem, please do not hesitate to contact me by any of the following methods. Thank you very much.

Round 3
Reviewer 1 Report
In this study a magnesium alloy is forged and then extruded with a large extrusion ratio. The resulting increase in the strength and the microstructures in the forged and the extruded state are compared. This includes study of texture.
There are some weaknesses in the study or the manuscript. An increase in the strength is expected, mainly due to the expected reduction in the grain size. The reduction in the grain size is not large, which will be expected of the large extrusion ratio employed here. It may be due to the fact that the extrusion temperature was relatively high. Secondly, the microstructure is not 'clean' enough, i.e., not fully recrystallized. Therefore any conclusions based on it are weak.
Second weakness is that the microstructure studies are not detailed (if that is the focus of the study). The phases are identified, correctly, but their distribution and volume are not known. Not that it would matter much, because the grain size effect will be dominant on the strength, and there cannot be expected to be much change in the phases.
Yet another weakness, most important, is that the explanations are hand waving. It is not explained why the particular Hall-Petch parameters were chosen. Are they for the same alloy after similar processing? It is full of loose or meaningless statements such as
"causing some grains rotate to certain orientations favorable or unfavorable for the activation of basal slips during tensile test," "some big and small grains,"
"some lamellae are parallel to each other within individual grain" (can they be different, especially if they are on basal planes?)
The entire Discussion section does not has much meaning, because it is not based on data. For example, how do we know that " the distribution of the β′ phase is much denser than that in the forged alloy" from Fig. 3?
The whole paragraph "Therefore, the dispersed and nanoscale β′ phase greatly contributes to the strength improvement of the extruded alloy. In addition, the LPSO phase is another important strengthening phase in the Mg-RE alloy due to its special lamellar morphology obstructing the sliding of dislocations. Densities of the LPSO phases precipitate on the matrix and kink in the extruded alloy suggestssever plastic deformation during the extrusion process. The kinking can release plastic deformation and avoid stress concentration in local areas, resulting in relatively uniform deformation and better mechanical properties of the extruded alloy. Moreover, the appearance of amounts of kinking LPSO lamellae can refine grains more uniformly, which is beneficial to strengthening the extruded alloy. Hence, the comprehensive effects of the dispersed distribution of tiny β′ phase and LPSO phase contribute significantly to the improvement of the strength and ductility of the extruded alloy. " is meaningless because it makes very general statements not specific to the present study.
Section 4.3 is not very meaningful either. Recrystallized and unrecrystallized grains will have different textures.
Section 4.4: the role of twinning is too complex to comment on, especially for an alloy of mixed grain types.
I hope this criticism will have positive impact on the authors' future work and manuscripts.
Reviewer 2 Report
Now the reviewer thinks that the manuscript is acceptable for publication.